# Linear Codes from Two Weakly Regular Plateaued Balanced Functions

**DOI:** 10.3390/e25020369

**Published:** 2023-02-17

**Authors:** Shudi Yang, Tonghui Zhang, Ping Li

**Affiliations:** 1School of Mathematical Sciences, Qufu Normal University, Jining 273165, China; 2School of Computer Science, South China Normal University, Guangzhou 510631, China

**Keywords:** linear code, weight distribution, Walsh transform, plateaued balanced function, 94B15, 14G50, 11T23

## Abstract

Linear codes with a few weights have been extensively studied due to their wide applications in secret sharing schemes, strongly regular graphs, association schemes, and authentication codes. In this paper, we choose the defining sets from two distinct weakly regular plateaued balanced functions, based on a generic construction of linear codes. Then we construct a family of linear codes with at most five nonzero weights. Their minimality is also examined and the result shows that our codes are helpful in secret sharing schemes.

## 1. Introduction

Throughout this paper, we will denote by $F_{p}$ a finite field with *p* elements, where *p* is an odd prime. An [n,k,d] linear code *C* of length *n* over $F_{p}$ is a *k*-dimensional linear subspace of $F_{p}^{n}$ with Hamming distance *d*. The Hamming weight of a codeword $c=(c_{0},c_{1},\cdots ,c_{n-1})$ is defined by $\mathrm{wt}\left(c\right)=\#\left\{c_{i}\neq 0:0⩽i⩽n-1\right\}$. Let $A_{w}$ be the number of codewords with weight *w* in *C*. By the weight distribution of *C*, we mean the sequence $\left(1,A_{1},A_{2},\cdots ,A_{n}\right)$. The code *C* is called *t*-weight if the number of nonzero $A_{j}$ in the sequence $\left(A_{1},A_{2},\cdots ,A_{n}\right)$ equals *t*. The weight distribution contains important information about the codes including the capabilities of error detection and correction. In recent years, many interesting articles have been published on good linear codes [1,2,3,4,5,6,7,8,9,10,11,12,13]. Besides, many linear codes with a few weights have been constructed from certain special functions, such as square functions [14], Boolean functions [1] and bent functions [8,15,16,17]. Among them, the plateaued functions, introduced by Zheng et al. in [18], have become one of the most attractive functions recently. The authors in [7,9,10,19] have given several families of linear codes using various weakly regular plateaued functions.

There are several methods to construct linear codes, and one of them goes back to the work of Ding et al. [20]. Let $q=p^{m}$ for a positive integer *m*, 0⩽s,t⩽m and $D=\left\{d_{1},d_{2},\cdots ,d_{n}\right\}\subseteq F_{q}^{*}$. A class of linear codes over $F_{p}$ is defined by
$$\begin{array}{c} C_{D}=\left\{c\left(a\right)=\left(Tr\left(ad_{1}\right),Tr\left(ad_{2}\right),\cdots ,Tr\left(ad_{n}\right)\right):a\in F_{q}\right\}, \end{array}$$
where Tr is the trace function from $F_{q}$ to $F_{p}$ defined by $Tr\left(x\right)=x+x^{p}+\cdots +x^{p^{m-1}}$ for $x\in F_{q}$. Here *D* is called the defining set of $C_{D}$. Many good linear codes have been derived from this generic approach [3,4,5,9,16,21]. For instance, Sınak et al. [9] constructed a family of linear codes by the defining set:$$\begin{array}{c} D=\left\{x\in F_{q}^{*}:f\left(x\right)=c\right\},c\in F_{p}, \end{array}$$
where *f* is a weakly regular *s*-plateaued balanced function.

As one of the generalizations of [20], Li et al. [22] defined a *p*-ary linear code by
(1)$$\begin{array}{c} C_{D}=\left\{c\left(a,b\right)={\left(Tr(ax+by)\right)}_{(x,y)\in D}:a,b\in F_{q}\right\}, \end{array}$$
where $D\subseteq F_{q}^{2}$ is also called a defining set. Based on this method, the authors in [2,6,10,11,12,13,17] constructed various linear codes from distinct defining sets. In particular, Cheng et al. in [2] introduced several linear codes of (Equation 1) with a few weights by the defining sets:(2)$$\begin{array}{cc} \; & D=\left\{\left(x,y\right)\in F_{q}^{2}\backslash \left\{\left(0,0\right)\right\}:f\left(x\right)+g\left(y\right)=0\right\},\\ \; & D_{SQ}=\left\{\left(x,y\right)\in F_{q}^{2}\backslash \left\{\left(0,0\right)\right\}:f\left(x\right)+g\left(y\right)\in SQ\right\},\\ \; & D_{NSQ}=\left\{\left(x,y\right)\in F_{q}^{2}\backslash \left\{\left(0,0\right)\right\}:f\left(x\right)+g\left(y\right)\in NSQ\right\}, \end{array}$$
where *f* and *g* are weakly regular *s*-plateaued unbalanced functions, SQ (resp. NSQ) represents the set of square (resp. non-square) elements in $F_{p}^{*}$. Later, Sınak et al. [10] constructed new linear codes from an extended defining set *D* of (Equation 2) by considering *f* and *g* to be weakly regular *s*-plateaued and *t*-plateaued functions, respectively. Inspired by the idea in [2,10], we choose a new defining set
$$\begin{array}{c} D_{f,g}=\left\{\left(x,y\right)\in F_{q}^{2}\backslash \left\{\left(0,0\right)\right\}:f\left(x\right)+g\left(y\right)=c\right\},c\in F_{p}^{*}, \end{array}$$
where *f* and *g* are weakly regular *s*-plateaued and *t*-plateaued balanced functions, respectively. In this paper, we will investigate the linear codes $C_{D_{f,g}}$ of (Equation 1) and determine their parameters and weight distributions using Walsh transform. In fact, as a generalization of [9], the codes we construct will partially extend the results of [2,10].

The rest of this paper is arranged as follows. A summary of weakly regular plateaued functions is presented in Section 2. Section 3 introduces some exponential sums, which will be employed in the subsequent sections. The main results about linear codes $C_{D_{f,g}}$ are given in Section 4, where we investigate the weight distributions of the codes. Section 5 illustrates the minimality and applications of these codes. Finally, in Section 6 we conclude the paper.

## 2. Mathematical Foundation

In this section, we will introduce some necessary tools about cyclotomic fields and weakly regular plateaued functions. Firstly, some notations are fixed.

(1)$q=p^{m}$, where *p* is an odd prime and *m* is a positive integer;(2)SQ (resp. NSQ) represents the set of square (resp. non-square) elements in $F_{p}^{*}$;(3)$\zeta_{p}$ is a primitive *p*-th root of unity;(4)Tr is the trace function from $F_{q}$ to $F_{p}$;(5)η is the quadratic character of $F_{p}^{*}$;(6)$p^{*}=\eta \left(-1\right)p={\left(-1\right)}^{\frac{p-1}{2}}p$ and hence $p^{m}=\eta^{m}\left(-1\right){\sqrt{p^{*}}}^{2m}$.

### 2.1. Cyclotomic Fields

A cyclotomic field $K=Q(\zeta_{p})$ is established from the rational field Q by adjoining $\zeta_{p}$. We call *K* the *p*-th cyclotomic field over Q. Actually, the field *K* is the splitting field of $x^{p}-1$, and *K* is a simple algebraic extension of Q as stated in Theorem 2.47 of [23]. We employ K/Q to stand for the field extension of *K* over Q.

**Lemma 1** ([24]). *Let K be the p-th cyclotomic field over Q. Then we have the following results.*


(1)
*The ring of integers in K is $Z[\zeta_{p}]$, where Z is the ring of integers, and $\left\{\zeta_{p}^{i}:1⩽i⩽p-1\right\}$ is an integer basis of $Z[\zeta_{p}]$.*
(2)
*The field extension K/Q is Galois of degree p−1, and the Galois group $Gal\left(K/Q\right)=\left\{\sigma_{z}:z\in F_{p}^{*}\right\}$, where the automorphism $\sigma_{z}$ of K is defined as $\sigma_{z}\left(\zeta_{p}\right)=\zeta_{p}^{z}$.*
(3)
*The cyclotomic field K has a unique quadratic subfield $Q(\sqrt{p^{*}})$. For $z\in F_{p}^{*}$, $\sigma_{z}\left(\sqrt{p^{*}}\right)=\eta \left(z\right)\sqrt{p^{*}}$.*



By Lemma 1, for any $z\in F_{p}^{*}$ and $x\in F_{p}$, we have $\sigma_{z}\left(\zeta_{p}^{x}\right)=\zeta_{p}^{zx}$ and $\sigma_{z}\left({\sqrt{p^{*}}}^{m}\right)=\eta^{m}\left(z\right){\sqrt{p^{*}}}^{m}$.

### 2.2. Weakly Regular Plateaued Functions

In this subsection, we will introduce some properties of weakly regular functions. Let *f* be a *p*-ary function from $F_{q}$ to $F_{p}$. The Walsh transform (see Page 73 of [25]) of *f* on $\beta \in F_{q}$ is defined as a complex-valued function ${\hat{\chi }}_{f}$ on $F_{q}$,
$$\begin{array}{c} {\hat{\chi }}_{f}\left(\beta \right)=\sum_{x\in F_{q}}\zeta_{p}^{f(x)-Tr(\beta x)}. \end{array}$$

A function *f* is said to be balanced over $F_{q}$ if *f* takes every element of $F_{p}$ the same number $p^{m-1}$ of pre-images. Otherwise, it is unbalanced. Clearly, *f* is balanced if and only if ${\hat{\chi }}_{f}\left(0\right)=0$.

Bent functions are the ones satisfying $|{\hat{\chi }}_{f}{\left(\beta \right)|}^{2}=p^{m}$. For a bent function *f*, if $p^{-\frac{m}{2}}{\hat{\chi }}_{f}\left(\beta \right)=\zeta_{p}^{g(\beta )}$ for every $\beta \in F_{q}$ and some *p*-ary function *g*, then *f* is called regular bent. On the other hand, *f* is weakly regular bent if there exists a complex number *u* with |u|=1 and a *p*-ary function *g* such that $up^{-\frac{m}{2}}{\hat{\chi }}_{f}\left(\beta \right)=\zeta_{p}^{g(\beta )}$ for all $\beta \in F_{q}$. The function *g* is also weakly regular bent.

As an extension of bent functions, Zheng et al. [18] firstly introduced the notion of plateaued functions in characteristic 2. It was later extended again by Mesnager [15] in any odd characteristic *p*. A function *f* is called *s*-plateaued if $|{\hat{\chi }}_{f}{\left(\beta \right)|}^{2}\in \left\{0,p^{m+s}\right\}$ for every $\beta \in F_{q}$, where *s* is an integer with 0⩽s⩽m. It is worth noting that every bent function is 0-plateaued. The Walsh support of an *s*-plateaued *f* is defined by
$$\begin{array}{c} S_{f}=\{\beta \in F_{q}:|{\hat{\chi }}_{f}\left(\beta \right){|}^{2}=p^{m+s}\}. \end{array}$$

By the Parseval identity, we have $\#S_{f}=p^{m-s}$, which verifies the following lemma.

**Lemma 2** (Lemma 1, [15]). *Let f be an s-plateaued function. Then for $\beta \in F_{q}$, $|{\hat{\chi }}_{f}{\left(\beta \right)|}^{2}$ takes $p^{m-s}$ times the value $p^{m+s}$ and $p^{m}-p^{m-s}$ times the value 0.*

The notion of weakly regular *s*-plateaued functions is due to Mesnager et al. [19].

**Definition 1** ([19]). *Let f be an s-plateaued function, where 0⩽s⩽m. Then, f is called weakly regular s-plateaued if there exists a complex number u with |u|=1, such that*
$$\begin{array}{c} {\hat{\chi }}_{f}\left(\beta \right)\in \left\{0,up^{\frac{m+s}{2}}\zeta_{p}^{g(\beta )}\right\} \end{array}$$
*for all $\beta \in F_{q}$, with g being a p-ary function over $F_{q}$ and g(β)=0 for all $\beta \in F_{q}\backslash S_{f}$. Otherwise, f is called non-weakly regular s-plateaued. Note that a weakly regular f is said to be regular if u=1. Moreover, if a weakly regular s-plateaued function f satisfies ${\hat{\chi }}_{f}\left(0\right)=0$ (resp. ${\hat{\chi }}_{f}\left(0\right)\neq 0$ ), then f is said to be weakly regular s-plateaued balanced (resp. unbalanced).*

**Lemma 3** (Lemma 5, [19]). *Let $\beta \in F_{q}$ and f be a weakly regular s-plateaued function. For every $\beta \in S_{f}$ we have*
$$\begin{array}{c} {\hat{\chi }}_{f}\left(\beta \right)=\varepsilon_{f}{\sqrt{p^{*}}}^{m+s}\zeta_{p}^{f^{*}\left(\beta \right)}, \end{array}$$*where $\varepsilon_{f}=\pm 1$ is the sign of ${\hat{\chi }}_{f}$ and $f^{*}$ is a p-ary function over $F_{q}$ with $f^{*}\left(\beta \right)=0$ for all $\beta \in F_{q}\backslash S_{f}$.*

In the literature, two subclasses of weakly regular plateaued functions were introduced by setting two homogeneous conditions. Let *f* be a weakly regular *s*-plateaued function, where 0⩽s⩽m, and let WRPB (resp. WRP) denote the class of these balanced (resp. unbalanced) functions that meet the following two homogeneous conditions:(1)f(0)=0;(2)There exists an even positive integer $h_{f}$ with $\gcd(h_{f}-1,p-1)=1$, such that $f\left(zx\right)=z^{h_{f}}f\left(x\right)$ for any $z\in F_{p}^{*}$ and $x\in F_{q}$.

**Remark 1.** *For every f ∈* WRPB (resp. f ∈ WRP), we have $0\notin S_{f}$ (resp. $0\in S_{f}$).

The classical work on weakly regular *s*-plateaued functions was presented by Mesnager et al. [7] and Sınak [9].

**Lemma 4** (Lemma 6, [7] and Lemma 4, [9]). *Let $\beta \in F_{q}$ and f∈WRPB or f∈WRP with ${\hat{\chi }}_{f}\left(\beta \right)=\varepsilon_{f}{\sqrt{p^{*}}}^{m+s}\zeta_{p}^{f^{*}\left(\beta \right)}$. Then, for every $z\in F_{p}^{*}$, $z\beta \in S_{f}$ if $\beta \in S_{f}$, otherwise, $z\beta \in F_{q}\backslash S_{f}$.*

**Lemma 5** (Propositions 2 and 3, [7] and Lemma 5, [9]). *Let $\beta \in F_{q}$ and f∈WRPB or f∈WRP with ${\hat{\chi }}_{f}\left(\beta \right)=\varepsilon_{f}{\sqrt{p^{*}}}^{m+s}\zeta_{p}^{f^{*}\left(\beta \right)}$ for every $\beta \in S_{f}$. Then*

(1)
*$f^{*}\left(0\right)=0$;*
(2)
*We have $f^{*}\left(z\beta \right)=z^{l_{f}}f^{*}\left(\beta \right)$ for any $z\in F_{p}^{*}$ and $\beta \in S_{f}$, where $l_{f}$ is an even positive integer with $\gcd(l_{f}-1,p-1)=1$.*


**Lemma 6** (Lemma 4, [7]). *Let f be a weakly regular s-plateaued function. Then for $x\in F_{q}$,*
$$\begin{array}{c} \sum_{\beta \in S_{f}}\zeta_{p}^{f^{*}\left(\beta \right)+Tr\left(\beta x\right)}=\varepsilon_{f}\eta^{m}\left(-1\right){\sqrt{p^{*}}}^{m-s}\zeta_{p}^{f(x)}, \end{array}$$
*where $\varepsilon_{f}=\pm 1$ is the sign of ${\hat{\chi }}_{f}$ and $f^{*}$ is a p-ary function over $F_{q}$ with $f^{*}\left(\beta \right)=0$ for all $\beta \in F_{q}\backslash S_{f}$.*

**Lemma 7** (Lemma 10, [7]). *Let f be a weakly regular s-plateaued function with ${\hat{\chi }}_{f}\left(\beta \right)=\varepsilon_{f}{\sqrt{p^{*}}}^{m+s}\zeta_{p}^{f^{*}\left(\beta \right)}$ for every $\beta \in S_{f}$. For $c\in F_{p}$, define*
$$\begin{array}{c} N_{f}\left(c\right)=\#\left\{\beta \in S_{f}:f^{*}\left(\beta \right)=c\right\}. \end{array}$$
*When m−s is even, we have*

$$\begin{array}{c} N_{f}\left(c\right)=\left\{\begin{array}{cc} p^{m-s-1}+\varepsilon_{f}\eta^{m+1}\left(-1\right)\left(p-1\right){\sqrt{p^{*}}}^{m-s-2}, & if\;c=0,\\ p^{m-s-1}-\varepsilon_{f}\eta^{m+1}\left(-1\right){\sqrt{p^{*}}}^{m-s-2}, & if\;c\in F_{p}^{*}. \end{array}\right. \end{array}$$


*Otherwise,*

$$\begin{array}{c} N_{f}\left(c\right)=\left\{\begin{array}{cc} p^{m-s-1}, & if\;c=0,\\ p^{m-s-1}+\varepsilon_{f}\eta \left(c\right)\eta^{m}\left(-1\right){\sqrt{p^{*}}}^{m-s-1}, & if\;c\in F_{p}^{*}. \end{array}\right. \end{array}$$



## 3. Exponential Sums Associated with Functions in WRPB

In this section, we will present auxiliary results on exponential sums related to weakly regular plateaued balanced functions. These results are very useful in the subsequent sections.

**Lemma 8** ([23]). *For any $b\in F_{p}^{*}$, we have*

(1)

$\sum_{x\in F_{p}^{*}}\zeta_{p}^{bx}=-1;$

(2)

$\sum_{x\in F_{p}^{*}}\eta \left(x\right)=0;$

(3)

$\sum_{x\in F_{p}^{*}}\eta \left(x\right)\zeta_{p}^{x}=\sqrt{p^{*}};$

(4)
*$\sum_{x\in F_{p}}\zeta_{p}^{bx^{2}}=\eta \left(b\right)\sqrt{p^{*}}$.*


**Lemma 9.** 
*Assume that g∈WRPB or g∈WRP with ${\hat{\chi }}_{g}\left(\beta \right)=\varepsilon_{g}{\sqrt{p^{*}}}^{m+t}\zeta_{p}^{g^{*}\left(\beta \right)}$ for every $\beta \in S_{g}$. For $c\in F_{p}^{*}$, we define*

$$\begin{array}{cc} R_{g,SQ}\left(c\right) & =\#\{b\in S_{g}:g^{*}\left(b\right)-c\in SQ\},\\ R_{g,NSQ}\left(c\right) & =\#\{b\in S_{g}:g^{*}\left(b\right)-c\in NSQ\}. \end{array}$$


*When m−t is even, we have*

$$\begin{array}{cc} R_{g,SQ}\left(c\right) & =\frac{p-1}{2}p^{m-t-1}+\frac{1+p\eta (-c)}{2}\varepsilon_{g}\eta^{m+1}\left(-1\right){\sqrt{p^{*}}}^{m-t-2},\\ R_{g,NSQ}\left(c\right) & =\frac{p-1}{2}p^{m-t-1}+\frac{1-p\eta (-c)}{2}\varepsilon_{g}\eta^{m+1}\left(-1\right){\sqrt{p^{*}}}^{m-t-2}. \end{array}$$


*Otherwise, we have*

$$\begin{array}{cc} R_{g,SQ}\left(c\right) & =\frac{p-1}{2}p^{m-t-1}-\frac{1+\eta (c)}{2}\varepsilon_{g}\eta^{m}\left(-1\right){\sqrt{p^{*}}}^{m-t-1},\\ R_{g,NSQ}\left(c\right) & =\frac{p-1}{2}p^{m-t-1}+\frac{1-\eta (c)}{2}\varepsilon_{g}\eta^{m}\left(-1\right){\sqrt{p^{*}}}^{m-t-1}. \end{array}$$



**Proof.** For $c\in F_{p}^{*}$, we define
$$\begin{array}{c} R=\sum_{z\in F_{p}}\sum_{b\in S_{g}}\zeta_{p}^{z^{2}\left(g^{*}\left(b\right)-c\right)}. \end{array}$$Note that
$$\begin{array}{c} N_{g}\left(c\right)=\#\left\{b\in S_{g}:g^{*}\left(b\right)=c\right\}. \end{array}$$From Lemmas 2 and 8, we obtain
(3)$$\begin{array}{cc} \; & N_{g}\left(c\right)+R_{g,SQ}\left(c\right)+R_{g,NSQ}\left(c\right)=p^{m-t}, \end{array}$$
(4)$$\begin{array}{cc} \; & R=pN_{g}\left(c\right)+\sqrt{p^{*}}R_{g,SQ}\left(c\right)-\sqrt{p^{*}}R_{g,NSQ}\left(c\right). \end{array}$$On the other hand, from Lemmas 6 and 8, we have
(5)$$\begin{array}{cc} R & =\sum_{b\in S_{g}}\left(1+\sum_{z\in F_{p}^{*}}\zeta_{p}^{z^{2}\left(g^{*}\left(b\right)-c\right)}\right)\\ \; & =p^{m-t}+\sum_{z\in F_{p}^{*}}\zeta_{p}^{-cz^{2}}\sigma_{z^{2}}\left(\sum_{b\in S_{g}}\zeta_{p}^{g^{*}\left(b\right)}\right)\\ \; & =p^{m-t}+\varepsilon_{g}\eta^{m}\left(-1\right){\sqrt{p^{*}}}^{m-t}\sum_{z\in F_{p}^{*}}\zeta_{p}^{-cz^{2}}\\ \; & =p^{m-t}+\varepsilon_{g}\eta^{m}\left(-1\right){\sqrt{p^{*}}}^{m-t}\left(\eta \left(-c\right)\sqrt{p^{*}}-1\right). \end{array}$$The desired assertion then follows from (Equation 3)–(Equation 5) and Lemma 7. □

**Lemma 10** (Lemma 3.12, [10]). *Assume f,g∈WRPB or f,g∈WRP with ${\hat{\chi }}_{f}\left(\alpha \right)=\varepsilon_{f}{\sqrt{p^{*}}}^{m+s}\zeta_{p}^{f^{*}\left(\alpha \right)}$ and ${\hat{\chi }}_{g}\left(\beta \right)=\varepsilon_{g}{\sqrt{p^{*}}}^{m+t}\zeta_{p}^{g^{*}\left(\beta \right)}$ for every $\alpha \in S_{f}$ and every $\beta \in S_{g}$, respectively. Let*
$$\begin{array}{cc} \; & T\left(0\right)=\#\{\left(a,b\right)\in S_{f}\times S_{g}:f^{*}\left(a\right)+g^{*}\left(b\right)=0\},\\ \; & T\left(c\right)=\#\left\{\left(a,b\right)\in S_{f}\times S_{g}:f^{*}\left(a\right)+g^{*}\left(b\right)=c\right\},where\;c\in F_{p}^{*}. \end{array}$$
*Then we have*

$$\begin{array}{cc} \; & T\left(0\right)=\left\{\begin{array}{cc} p^{2m-s-t-1}+\frac{p-1}{p}\varepsilon_{f}\varepsilon_{g}{\sqrt{p^{*}}}^{2m-s-t}, & if\;s+t\;is\;even,\\ p^{2m-s-t-1}, & if\;s+t\;is\;odd, \end{array}\right.\\ \; & T\left(c\right)=\left\{\begin{array}{cc} p^{2m-s-t-1}-p^{-1}\varepsilon_{f}\varepsilon_{g}{\sqrt{p^{*}}}^{2m-s-t}, & if\;s+t\;is\;even,\\ p^{2m-s-t-1}+\eta \left(c\right)\varepsilon_{f}\varepsilon_{g}{\sqrt{p^{*}}}^{2m-s-t-1}, & if\;s+t\;is\;odd. \end{array}\right. \end{array}$$



**Lemma 11.** 
*Assume f,g∈WRPB or f,g∈WRP with ${\hat{\chi }}_{f}\left(\alpha \right)=\varepsilon_{f}{\sqrt{p^{*}}}^{m+s}\zeta_{p}^{f^{*}\left(\alpha \right)}$ and ${\hat{\chi }}_{g}\left(\beta \right)=\varepsilon_{g}{\sqrt{p^{*}}}^{m+t}\zeta_{p}^{g^{*}\left(\beta \right)}$ for every $\alpha \in S_{f}$ and every $\beta \in S_{g}$, respectively. For $c\in F_{p}^{*}$, define*

$$\begin{array}{cc} T_{SQ}\left(c\right) & =\#\{\left(a,b\right)\in S_{f}\times S_{g}:\frac{f^{*}\left(a\right)+g^{*}\left(b\right)}{c}\in SQ\},\\ T_{NSQ}\left(c\right) & =\#\{\left(a,b\right)\in S_{f}\times S_{g}:\frac{f^{*}\left(a\right)+g^{*}\left(b\right)}{c}\in NSQ\}. \end{array}$$


*When s+t is even, we have*

$$\begin{array}{c} T_{SQ}\left(c\right)=T_{NSQ}\left(c\right)=\frac{p-1}{2}\left(p^{2m-s-t-1}-p^{-1}\varepsilon_{f}\varepsilon_{g}{\sqrt{p^{*}}}^{2m-s-t}\right). \end{array}$$


*Otherwise,*

$$\begin{array}{cc} T_{SQ}\left(c\right) & =\frac{p-1}{2}\left(p^{2m-s-t-1}+\varepsilon_{f}\varepsilon_{g}\eta \left(c\right){\sqrt{p^{*}}}^{2m-s-t-1}\right),\\ T_{NSQ}\left(c\right) & =\frac{p-1}{2}\left(p^{2m-s-t-1}-\varepsilon_{f}\varepsilon_{g}\eta \left(c\right){\sqrt{p^{*}}}^{2m-s-t-1}\right). \end{array}$$



**Proof.** Let $c\in F_{p}^{*}$. It is obvious that
$$\begin{array}{c} \#\{\left(a,b\right)\in S_{f}\times S_{g}:\frac{f^{*}\left(a\right)+g^{*}\left(b\right)}{c}=0\}=T\left(0\right). \end{array}$$We define an exponential sum
$$\begin{array}{c} T=\sum_{z\in F_{p}}\sum_{a\in S_{f}}\sum_{b\in S_{g}}\zeta_{p}^{z^{2}\frac{f^{*}\left(a\right)+g^{*}\left(b\right)}{c}}. \end{array}$$It is evident by definition that
$$\begin{array}{c} T=pT\left(0\right)+\sqrt{p^{*}}T_{SQ}\left(c\right)-\sqrt{p^{*}}T_{NSQ}\left(c\right). \end{array}$$On the other hand, it follows from Lemmas 6 and 8 that
$$\begin{array}{cc} T & =\sum_{z\in F_{p}}\sum_{a\in S_{f}}\sum_{b\in S_{g}}\zeta_{p}^{z^{2}\frac{f^{*}\left(a\right)+g^{*}\left(b\right)}{c}}\\ \; & =\sum_{a\in S_{f}}\sum_{b\in S_{g}}\left(\sum_{z\in F_{p}^{*}}\zeta_{p}^{\frac{z^{2}}{c}\left(f^{*}\left(a\right)+g^{*}\left(b\right)\right)}+1\right)\\ \; & =p^{2m-s-t}+\sum_{z\in F_{p}^{*}}\sigma_{\frac{z^{2}}{c}}\left(\sum_{a\in S_{f}}\zeta_{p}^{f^{*}\left(a\right)}\sum_{b\in S_{g}}\zeta_{p}^{g^{*}\left(b\right)}\right)\\ \; & =p^{2m-s-t}+\varepsilon_{f}\varepsilon_{g}{\sqrt{p^{*}}}^{2m-s-t}\sum_{z\in F_{p}^{*}}\eta^{2m-s-t}\left(\frac{z^{2}}{c}\right)\\ \; & =\left\{\begin{array}{cc} p^{2m-s-t}+\left(p-1\right)\varepsilon_{f}\varepsilon_{g}{\sqrt{p^{*}}}^{2m-s-t}, & if\;s+t\;is\;even,\\ p^{2m-s-t}+\left(p-1\right)\eta \left(c\right)\varepsilon_{f}\varepsilon_{g}{\sqrt{p^{*}}}^{2m-s-t}, & if\;s+t\;is\;odd. \end{array}\right. \end{array}$$Combining Lemma 10 and the fact that
$$\begin{array}{c} T\left(0\right)+T_{SQ}\left(c\right)+T_{NSQ}\left(c\right)=\#\left\{\left(a,b\right)\in S_{f}\times S_{g}\right\}=p^{2m-s-t}, \end{array}$$
we obtain the desired assertion. □

**Lemma 12.** 
*Assume f,g∈WRPB or f,g∈WRP with ${\hat{\chi }}_{f}\left(\alpha \right)=\varepsilon_{f}{\sqrt{p^{*}}}^{m+s}\zeta_{p}^{f^{*}\left(\alpha \right)}$ and ${\hat{\chi }}_{g}\left(\beta \right)=\varepsilon_{g}{\sqrt{p^{*}}}^{m+t}\zeta_{p}^{g^{*}\left(\beta \right)}$ for every $\alpha \in S_{f}$ and every $\beta \in S_{g}$, respectively. For $c\in F_{p}^{*}$, define*

$$\begin{array}{cc} V_{SQ}\left(c\right) & =\#\{\left(a,b\right)\in S_{f}\times S_{g}:f^{*}\left(a\right)+g^{*}\left(b\right)-c\in SQ\},\\ V_{NSQ}\left(c\right) & =\#\{\left(a,b\right)\in S_{f}\times S_{g}:f^{*}\left(a\right)+g^{*}\left(b\right)-c\in NSQ\}. \end{array}$$


*When s+t is even, we get*

$$\begin{array}{cc} V_{SQ}\left(c\right) & =\frac{p-1}{2}p^{2m-s-t-1}+\frac{1+p\eta (-c)}{2p}\varepsilon_{f}\varepsilon_{g}{\sqrt{p^{*}}}^{2m-s-t},\\ V_{NSQ}\left(c\right) & =\frac{p-1}{2}p^{2m-s-t-1}+\frac{1-p\eta (-c)}{2p}\varepsilon_{f}\varepsilon_{g}{\sqrt{p^{*}}}^{2m-s-t}. \end{array}$$


*Otherwise,*

$$\begin{array}{cc} V_{SQ}\left(c\right) & =\frac{p-1}{2}p^{2m-s-t-1}-\frac{1+\eta (c)}{2}\varepsilon_{f}\varepsilon_{g}{\sqrt{p^{*}}}^{2m-s-t-1},\\ V_{NSQ}\left(c\right) & =\frac{p-1}{2}p^{2m-s-t-1}+\frac{1-\eta (c)}{2}\varepsilon_{f}\varepsilon_{g}{\sqrt{p^{*}}}^{2m-s-t-1}. \end{array}$$



**Proof.** Obviously,
$$\begin{array}{c} \#\{\left(a,b\right)\in S_{f}\times S_{g}:f^{*}\left(a\right)+g^{*}\left(b\right)-c=0\}=T\left(c\right). \end{array}$$Let us define an exponential sum
$$\begin{array}{c} V=\sum_{z\in F_{p}}\sum_{a\in S_{f}}\sum_{b\in S_{g}}\zeta_{p}^{z^{2}\left(f^{*}\left(a\right)+g^{*}\left(b\right)-c\right)}. \end{array}$$Clearly,
$$\begin{array}{c} V=pT\left(c\right)+\sqrt{p^{*}}V_{SQ}\left(c\right)-\sqrt{p^{*}}V_{NSQ}\left(c\right). \end{array}$$By a similar procedure as we have done in the proof of Lemma 11, we have
$$\begin{array}{cc} V & =\sum_{z\in F_{p}}\sum_{a\in S_{f}}\sum_{b\in S_{g}}\zeta_{p}^{z^{2}\left(f^{*}\left(a\right)+g^{*}\left(b\right)-c\right)}\\ \; & =\sum_{a\in S_{f}}\sum_{b\in S_{g}}\left(\sum_{z\in F_{p}^{*}}\zeta_{p}^{z^{2}\left(f^{*}\left(a\right)+g^{*}\left(b\right)-c\right)}+1\right)\\ \; & =p^{2m-s-t}+\sum_{z\in F_{p}^{*}}\zeta_{p}^{-cz^{2}}\sigma_{z^{2}}\left(\sum_{a\in S_{f}}\zeta_{p}^{f^{*}\left(a\right)}\sum_{b\in S_{g}}\zeta_{p}^{g^{*}\left(b\right)}\right)\\ \; & =p^{2m-s-t}+\varepsilon_{f}\varepsilon_{g}{\sqrt{p^{*}}}^{2m-s-t}\sum_{z\in F_{p}^{*}}\zeta_{p}^{-cz^{2}}\\ \; & =p^{2m-s-t}+\varepsilon_{f}\varepsilon_{g}{\sqrt{p^{*}}}^{2m-s-t}\left(\eta \left(-c\right)\sqrt{p^{*}}-1\right). \end{array}$$Using Lemma 10 and the fact that
$$\begin{array}{c} T\left(c\right)+V_{SQ}\left(c\right)+V_{NSQ}\left(c\right)=\#\left\{\left(a,b\right)\in S_{f}\times S_{g}\right\}=p^{2m-s-t}, \end{array}$$
we complete the proof of this lemma. □

## 4. Main Results

Before we go any further, we make the following assumptions for the remainder of the paper. Assume that f,g∈WRPB with ${\hat{\chi }}_{f}\left(\alpha \right)=\varepsilon_{f}{\sqrt{p^{*}}}^{m+s}\zeta_{p}^{f^{*}\left(\alpha \right)}$ and ${\hat{\chi }}_{g}\left(\beta \right)=\varepsilon_{g}{\sqrt{p^{*}}}^{m+t}\zeta_{p}^{g^{*}\left(\beta \right)}$, where $\varepsilon_{f},\varepsilon_{g}\in \left\{\pm 1\right\}$ and 0⩽s,t⩽m for every $\alpha \in S_{f}$ and every $\beta \in S_{g}$, respectively. Here $f^{*}$ and $g^{*}$ are defined by Lemma 5 satisfying $f^{*}\left(zx\right)=z^{l_{f}}f^{*}\left(x\right)$ and $g^{*}\left(zx\right)=z^{l_{g}}g^{*}\left(x\right)$, where $z\in F_{p}^{*}$, $x\in F_{q}$ and $l_{f},l_{g}\in \left\{2,p-1\right\}$. In order to determine the weight distributions of $C_{D_{f,g}}$, we define
(6)$$\begin{array}{c} N_{c}=\#\left\{\left(x,y\right)\in F_{q}^{2}\backslash \left\{\left(0,0\right)\right\}:f\left(x\right)+g\left(y\right)=c,Tr\left(ax+by\right)=0\right\}, \end{array}$$
where $\left(a,b\right)\in F_{q}^{2}\backslash \left\{\left(0,0\right)\right\}$ and $c\in F_{p}^{*}$.

### 4.1. The Determination of $N_{c}$

In fact, the value $N_{0}$ was investigated in [10]. Now we only dedicate ourselves to exploring the case that c≠0. We shall determine the values of $N_{c}$ of (Equation 6) for c≠0 in Lemmas 13 and 14. Without loss of generality, when $l_{f}\neq l_{g}$, we only consider the case that $l_{f}=2$ and $l_{g}=p-1$.

**Lemma 13.** 
*Suppose that s+t is even, (a,b)≠(0,0) and $c\in F_{p}^{*}$. Then, if $\left(a,b\right)\notin S_{f}\times S_{g}$, we always have $N_{c}=p^{2m-2}$, and if $\left(a,b\right)\in S_{f}\times S_{g}$, we have the following cases. When $l_{f}=l_{g}=p-1$,*

$$\begin{array}{c} N_{c}=\left\{\begin{array}{cc} p^{2m-2}+{\left(p-1\right)}^{2}\varepsilon_{f}\varepsilon_{g}{\sqrt{p^{*}}}^{2(m-2)+s+t}, & if\;f^{*}\left(a\right)+g^{*}\left(b\right)=c,\\ p^{2m-2}-\left(p-1\right)\varepsilon_{f}\varepsilon_{g}{\sqrt{p^{*}}}^{2(m-2)+s+t}, & if\;f^{*}\left(a\right)+g^{*}\left(b\right)\neq c. \end{array}\right. \end{array}$$


*When lf=lg=2,*

$$\begin{array}{c} N_{c}=\left\{\begin{array}{cc} p^{2m-2}+\left(p+1\right)\varepsilon_{f}\varepsilon_{g}{\sqrt{p^{*}}}^{2(m-2)+s+t}, & if\;\eta (f^{*}\left(a\right)+g^{*}\left(b\right))=\eta \left(c\right),\\ p^{2m-2}-\left(p-1\right)\varepsilon_{f}\varepsilon_{g}{\sqrt{p^{*}}}^{2(m-2)+s+t}, & \mathrm{otherwise}. \end{array}\right. \end{array}$$


*Otherwise, when lf=2 and lg=p−1,*

$$\begin{array}{c} N_{c}=\left\{\begin{array}{cc} p^{2m-2}+{\left(p-1\right)}^{2}\varepsilon_{f}\varepsilon_{g}{\sqrt{p^{*}}}^{2(m-2)+s+t}, & if\;f^{*}\left(a\right)=0,g^{*}\left(b\right)=c,\\ p^{2m-2}-\left(p-1\right)\varepsilon_{f}\varepsilon_{g}{\sqrt{p^{*}}}^{2(m-2)+s+t}, & if\;f^{*}\left(a\right)=0,g^{*}\left(b\right)\neq c\\ \; & or\;f^{*}\left(a\right)\neq 0,g^{*}\left(b\right)=c,\\ p^{2m-2}+\left(\eta \left(-1\right)p+1\right)\varepsilon_{f}\varepsilon_{g}{\sqrt{p^{*}}}^{2(m-2)+s+t}, & if\;f^{*}\left(a\right)\left(g^{*}\left(b\right)-c\right)\in SQ,\\ p^{2m-2}-\left(\eta \left(-1\right)p-1\right)\varepsilon_{f}\varepsilon_{g}{\sqrt{p^{*}}}^{2(m-2)+s+t}, & if\;f^{*}\left(a\right)\left(g^{*}\left(b\right)-c\right)\in NSQ. \end{array}\right. \end{array}$$



**Proof.** Let c≠0. By definition and the orthogonal property of characters,
(7)$$\begin{array}{cc} N_{c} & =\frac{1}{p^{2}}\sum_{x,y\in F_{q}}\sum_{z\in F_{p}}{\zeta_{p}}^{z(f(x)+g(y)-c)}\sum_{h\in F_{p}}{\zeta_{p}}^{hTr(ax+by)}-\frac{1}{p}\sum_{z\in F_{p}}{\zeta_{p}}^{-cz}\\ \; & =p^{2m-2}+\frac{1}{p^{2}}\sum_{x,y\in F_{q}}\sum_{z\in F_{p}^{*}}{\zeta_{p}}^{-cz}\sum_{h\in F_{p}^{*}}{\zeta_{p}}^{z(f(x)+g(y))+hTr(ax+by)}\\ \; & =p^{2m-2}+p^{-2}S_{c}, \end{array}$$
where
$$\begin{array}{c} S_{c}=\sum_{x,y\in F_{q}}\sum_{z\in F_{p}^{*}}{\zeta_{p}}^{-cz}\sum_{h\in F_{p}^{*}}{\zeta_{p}}^{z(f(x)+g(y))+hTr(ax+by)}. \end{array}$$Now let us determine $S_{c}$. It follows that
(8)$$\begin{array}{cc} S_{c} & =\sum_{z\in F_{p}^{*}}\zeta_{p}^{-cz}\sum_{h\in F_{p}^{*}}\sum_{x\in F_{q}}{\zeta_{p}}^{zf(x)-Tr(hax)}\sum_{y\in F_{q}}{\zeta_{p}}^{zg(y)-Tr(hby)}\\ \; & =\sum_{z\in F_{p}^{*}}\zeta_{p}^{-cz}\sum_{h\in F_{p}^{*}}\sum_{x\in F_{q}}{\zeta_{p}}^{z(f\left(x\right)-Tr\left(\frac{h}{z}ax\right))}\sum_{y\in F_{q}}{\zeta_{p}}^{z(g\left(y\right)-Tr\left(\frac{h}{z}by\right))}\\ \; & =\sum_{z\in F_{p}^{*}}\zeta_{p}^{-cz}\sum_{h\in F_{p}^{*}}\sigma_{z}\left({\hat{\chi }}_{f}\left(\frac{ha}{z}\right){\hat{\chi }}_{g}\left(\frac{hb}{z}\right)\right). \end{array}$$When $\left(a,b\right)\notin S_{f}\times S_{g}$, from Lemma 4, we deduce that $\left(\frac{ha}{z},\frac{hb}{z}\right)\notin S_{f}\times S_{g}$ for $z,h\in F_{p}^{*}$. Then one easily checks
$$S_{c}=0.$$When $\left(a,b\right)\in S_{f}\times S_{g}$, again from Lemma 4, we see that $\left(\frac{ha}{z},\frac{hb}{z}\right)\in S_{f}\times S_{g}$ for $z,h\in F_{p}^{*}$. The valuation of $S_{c}$ is considered naturally under three cases of $l_{f}=l_{g}=p-1$, $l_{f}=l_{g}=2$ and $l_{f}\neq l_{g}$, respectively.

(1)The first case is that $l_{f}=l_{g}=p-1$. From Lemma 5,
$$\begin{array}{cc} S_{c} & =\varepsilon_{f}\varepsilon_{g}{\sqrt{p^{*}}}^{2m+s+t}\sum_{z\in F_{p}^{*}}\zeta_{p}^{-cz}\sum_{h\in F_{p}^{*}}\zeta_{p}^{z{\left(\frac{h}{z}\right)}^{p-1}\left(f^{*}\left(a\right)+g^{*}\left(b\right)\right)}\\ \; & =\varepsilon_{f}\varepsilon_{g}{\sqrt{p^{*}}}^{2m+s+t}\sum_{h\in F_{p}^{*}}\sum_{z\in F_{p}^{*}}\zeta_{p}^{(f^{*}\left(a\right)+g^{*}\left(b\right)-c)z}\\ \; & =\left\{\begin{array}{cc} {\left(p-1\right)}^{2}\varepsilon_{f}\varepsilon_{g}{\sqrt{p^{*}}}^{2m+s+t}, & if\;f^{*}\left(a\right)+g^{*}\left(b\right)=c,\\ -\left(p-1\right)\varepsilon_{f}\varepsilon_{g}{\sqrt{p^{*}}}^{2m+s+t}, & if\;f^{*}\left(a\right)+g^{*}\left(b\right)\neq c. \end{array}\right. \end{array}$$(2)The second case is that $l_{f}=l_{g}=2$. Again from Lemma 5,
$$\begin{array}{cc} S_{c} & =\varepsilon_{f}\varepsilon_{g}{\sqrt{p^{*}}}^{2m+s+t}\sum_{z\in F_{p}^{*}}\zeta_{p}^{-cz}\sum_{h\in F_{p}^{*}}\zeta_{p}^{z{\left(\frac{h}{z}\right)}^{2}\left(f^{*}\left(a\right)+g^{*}\left(b\right)\right)}\\ \; & =\varepsilon_{f}\varepsilon_{g}{\sqrt{p^{*}}}^{2m+s+t}\sum_{z\in F_{p}^{*}}\zeta_{p}^{-cz}\sum_{h\in F_{p}^{*}}\zeta_{p}^{\frac{f^{*}\left(a\right)+g^{*}\left(b\right)}{z}h^{2}}\\ \; & =\varepsilon_{f}\varepsilon_{g}{\sqrt{p^{*}}}^{2m+s+t}\sum_{z\in F_{p}^{*}}\zeta_{p}^{-cz}\left(\sum_{h\in F_{p}}\zeta_{p}^{\frac{f^{*}\left(a\right)+g^{*}\left(b\right)}{z}h^{2}}-1\right)\\ \; & =\left\{\begin{array}{cc} -\left(p-1\right)\varepsilon_{f}\varepsilon_{g}{\sqrt{p^{*}}}^{2m+s+t}, & if\;f^{*}\left(a\right)+g^{*}\left(b\right)=0\\ \varepsilon_{f}\varepsilon_{g}{\sqrt{p^{*}}}^{2m+s+t}\left(\eta \left(-c\right)\eta \left(f^{*}\left(a\right)+g^{*}\left(b\right)\right)p^{*}+1\right), & if\;f^{*}\left(a\right)+g^{*}\left(b\right)\neq 0 \end{array}\right.\\ \; & =\left\{\begin{array}{cc} \left(p+1\right)\varepsilon_{f}\varepsilon_{g}{\sqrt{p^{*}}}^{2m+s+t}, & if\;\eta (f^{*}\left(a\right)+g^{*}\left(b\right))=\eta \left(c\right),\\ -\left(p-1\right)\varepsilon_{f}\varepsilon_{g}{\sqrt{p^{*}}}^{2m+s+t}, & \mathrm{otherwise}. \end{array}\right. \end{array}$$(3)The last case is that $l_{f}=2$ and $l_{g}=p-1$. From Lemma 5,
$$\begin{array}{cc} S_{c} & =\varepsilon_{f}\varepsilon_{g}{\sqrt{p^{*}}}^{2m+s+t}\sum_{z\in F_{p}^{*}}\zeta_{p}^{-cz}\sum_{h\in F_{p}^{*}}\zeta_{p}^{z\left({\left(\frac{h}{z}\right)}^{2}f^{*}\left(a\right)+{\left(\frac{h}{z}\right)}^{p-1}g^{*}\left(b\right)\right)}\\ \; & =\varepsilon_{f}\varepsilon_{g}{\sqrt{p^{*}}}^{2m+s+t}\sum_{z\in F_{p}^{*}}\zeta_{p}^{z(g^{*}\left(b\right)-c)}\sum_{h\in F_{p}^{*}}\zeta_{p}^{\frac{h^{2}}{z}f^{*}\left(a\right)}\\ \; & =\varepsilon_{f}\varepsilon_{g}{\sqrt{p^{*}}}^{2m+s+t}\sum_{z\in F_{p}^{*}}\zeta_{p}^{z(g^{*}\left(b\right)-c)}\left(\sum_{h\in F_{p}}\zeta_{p}^{\frac{h^{2}}{z}f^{*}\left(a\right)}-1\right)\\ \; & =\left\{\begin{array}{cc} \left(p-1\right)\varepsilon_{f}\varepsilon_{g}{\sqrt{p^{*}}}^{2m+s+t}\sum_{z\in F_{p}^{*}}\zeta_{p}^{z(g^{*}\left(b\right)-c)}, & if\;f^{*}\left(a\right)=0\\ \varepsilon_{f}\varepsilon_{g}{\sqrt{p^{*}}}^{2m+s+t}\sum_{z\in F_{p}^{*}}\zeta_{p}^{z(g^{*}\left(b\right)-c)}\left(\eta \left(zf^{*}\left(a\right)\right)\sqrt{p^{*}}-1\right), & if\;f^{*}\left(a\right)\neq 0 \end{array}\right.\\ \; & =\left\{\begin{array}{cc} {\left(p-1\right)}^{2}\varepsilon_{f}\varepsilon_{g}{\sqrt{p^{*}}}^{2m+s+t}, & if\;f^{*}\left(a\right)=0,g^{*}\left(b\right)=c\\ -\left(p-1\right)\varepsilon_{f}\varepsilon_{g}{\sqrt{p^{*}}}^{2m+s+t}, & if\;f^{*}\left(a\right)=0,g^{*}\left(b\right)\neq c\\ -\left(p-1\right)\varepsilon_{f}\varepsilon_{g}{\sqrt{p^{*}}}^{2m+s+t}, & if\;f^{*}\left(a\right)\neq 0,g^{*}\left(b\right)=c\\ \varepsilon_{f}\varepsilon_{g}{\sqrt{p^{*}}}^{2m+s+t}\left(\eta \left(f^{*}\left(a\right)\left(g^{*}\left(b\right)-c\right)\right)p^{*}+1\right), & if\;f^{*}\left(a\right)\neq 0,g^{*}\left(b\right)\neq c \end{array}\right.\\ \; & =\left\{\begin{array}{cc} {\left(p-1\right)}^{2}\varepsilon_{f}\varepsilon_{g}{\sqrt{p^{*}}}^{2m+s+t}, & if\;f^{*}\left(a\right)=0,g^{*}\left(b\right)=c,\\ -\left(p-1\right)\varepsilon_{f}\varepsilon_{g}{\sqrt{p^{*}}}^{2m+s+t}, & if\;f^{*}\left(a\right)=0,g^{*}\left(b\right)\neq c,\\ -\left(p-1\right)\varepsilon_{f}\varepsilon_{g}{\sqrt{p^{*}}}^{2m+s+t}, & if\;f^{*}\left(a\right)\neq 0,g^{*}\left(b\right)=c,\\ \left(\eta \left(-1\right)p+1\right)\varepsilon_{f}\varepsilon_{g}{\sqrt{p^{*}}}^{2m+s+t}, & if\;f^{*}\left(a\right)\left(g^{*}\left(b\right)-c\right)\in SQ,\\ -\left(\eta \left(-1\right)p-1\right)\varepsilon_{f}\varepsilon_{g}{\sqrt{p^{*}}}^{2m+s+t}, & if\;f^{*}\left(a\right)\left(g^{*}\left(b\right)-c\right)\in NSQ. \end{array}\right. \end{array}$$

Hence, we obtain the desired assertion from (Equation 7). □

**Lemma 14.** 
*Suppose that s+t is odd, (a,b)≠(0,0) and $c\in F_{p}^{*}$. Then, if $\left(a,b\right)\notin S_{f}\times S_{g}$, we always have $N_{c}=p^{2m-2}$, and if $\left(a,b\right)\in S_{f}\times S_{g}$, we have the following cases. When $l_{f}=l_{g}=p-1$, we have*

$$\begin{array}{c} N_{c}=\left\{\begin{array}{cc} p^{2m-2}+\left(p-1\right)\varepsilon_{f}\varepsilon_{g}{\sqrt{p^{*}}}^{2m+s+t-3}, & if\;f^{*}\left(a\right)+g^{*}\left(b\right)-c\in SQ,\\ p^{2m-2}-\left(p-1\right)\varepsilon_{f}\varepsilon_{g}{\sqrt{p^{*}}}^{2m+s+t-3}, & if\;f^{*}\left(a\right)+g^{*}\left(b\right)-c\in NSQ,\\ p^{2m-2}, & \mathrm{otherwise} \end{array}\right. \end{array}$$


*When lf=lg=2,*

$$\begin{array}{c} N_{c}=\left\{\begin{array}{cc} p^{2m-2}+\eta \left(-c\right)\left(p-1\right)\varepsilon_{f}\varepsilon_{g}{\sqrt{p^{*}}}^{2m+s+t-3}, & if\;f^{*}\left(a\right)+g^{*}\left(b\right)=0,\\ p^{2m-2}-2\eta \left(-c\right)\varepsilon_{f}\varepsilon_{g}{\sqrt{p^{*}}}^{2m+s+t-3}, & if\;\eta (f^{*}\left(a\right)+g^{*}\left(b\right))=\eta \left(-c\right),\\ p^{2m-2}, & \mathrm{otherwise}. \end{array}\right. \end{array}$$


*Otherwise, when lf=2 and lg=p−1,*

$$\begin{array}{c} N_{c}=\left\{\begin{array}{cc} p^{2m-2}+\left(p-1\right)\varepsilon_{f}\varepsilon_{g}{\sqrt{p^{*}}}^{2m+s+t-3}, & if\;f^{*}\left(a\right)=0,g^{*}\left(b\right)-c\in SQ\\ \; & or\;f^{*}\left(a\right)\in SQ,g^{*}\left(b\right)=c,\\ p^{2m-2}-\left(p-1\right)\varepsilon_{f}\varepsilon_{g}{\sqrt{p^{*}}}^{2m+s+t-3}, & if\;f^{*}\left(a\right)=0,g^{*}\left(b\right)-c\in NSQ\\ \; & or\;f^{*}\left(a\right)\in NSQ,g^{*}\left(b\right)=c,\\ p^{2m-2}-2\varepsilon_{f}\varepsilon_{g}{\sqrt{p^{*}}}^{2m+s+t-3}, & if\;f^{*}\left(a\right)\in SQ,g^{*}\left(b\right)-c\in SQ,\\ p^{2m-2}+2\varepsilon_{f}\varepsilon_{g}{\sqrt{p^{*}}}^{2m+s+t-3}, & if\;f^{*}\left(a\right)\in NSQ,g^{*}\left(b\right)-c\in NSQ,\\ p^{2m-2}, & \mathrm{otherwise}. \end{array}\right. \end{array}$$



**Proof.** The proof is similar to that of Lemma 13 by noting (Equation 7) and (Equation 8). From (Equation 8), $S_{c}=0$ unless $\left(a,b\right)\in S_{f}\times S_{g}$. In the following, we set $\left(a,b\right)\in S_{f}\times S_{g}$.

(1) The first case we consider is that $l_{f}=l_{g}=p-1$. Then
$$\begin{array}{cc} S_{c} & =\varepsilon_{f}\varepsilon_{g}{\sqrt{p^{*}}}^{2m+s+t}\sum_{z\in F_{p}^{*}}\zeta_{p}^{-cz}\eta \left(z\right)\sum_{h\in F_{p}^{*}}\zeta_{p}^{z{\left(\frac{h}{z}\right)}^{p-1}\left(f^{*}\left(a\right)+g^{*}\left(b\right)\right)}\\ \; & =\varepsilon_{f}\varepsilon_{g}{\sqrt{p^{*}}}^{2m+s+t}\sum_{h\in F_{p}^{*}}\sum_{z\in F_{p}^{*}}\zeta_{p}^{(f^{*}\left(a\right)+g^{*}\left(b\right)-c)z}\eta \left(z\right)\\ \; & =\left\{\begin{array}{cc} 0, & if\;f^{*}\left(a\right)+g^{*}\left(b\right)=c\\ \eta \left(f^{*}\left(a\right)+g^{*}\left(b\right)-c\right)\left(p-1\right)\varepsilon_{f}\varepsilon_{g}{\sqrt{p^{*}}}^{2m+s+t+1}, & if\;f^{*}\left(a\right)+g^{*}\left(b\right)\neq c \end{array}\right.\\ \; & =\left\{\begin{array}{cc} 0, & if\;f^{*}\left(a\right)+g^{*}\left(b\right)=c,\\ \left(p-1\right)\varepsilon_{f}\varepsilon_{g}{\sqrt{p^{*}}}^{2m+s+t+1}, & if\;f^{*}\left(a\right)+g^{*}\left(b\right)-c\in SQ,\\ -\left(p-1\right)\varepsilon_{f}\varepsilon_{g}{\sqrt{p^{*}}}^{2m+s+t+1}, & if\;f^{*}\left(a\right)+g^{*}\left(b\right)-c\in NSQ. \end{array}\right. \end{array}$$

(2) The second case is that $l_{f}=l_{g}=2$. Now we have
$$\begin{array}{cc} S_{c} & =\varepsilon_{f}\varepsilon_{g}{\sqrt{p^{*}}}^{2m+s+t}\sum_{z\in F_{p}^{*}}\zeta_{p}^{-cz}\eta \left(z\right)\sum_{h\in F_{p}^{*}}\zeta_{p}^{z{\left(\frac{h}{z}\right)}^{2}\left(f^{*}\left(a\right)+g^{*}\left(b\right)\right)}\\ \; & =\varepsilon_{f}\varepsilon_{g}{\sqrt{p^{*}}}^{2m+s+t}\sum_{z\in F_{p}^{*}}\zeta_{p}^{-cz}\eta \left(z\right)\sum_{h\in F_{p}^{*}}\zeta_{p}^{\frac{f^{*}\left(a\right)+g^{*}\left(b\right)}{z}h^{2}}\\ \; & =\varepsilon_{f}\varepsilon_{g}{\sqrt{p^{*}}}^{2m+s+t}\sum_{z\in F_{p}^{*}}\zeta_{p}^{-cz}\eta \left(z\right)\left(\sum_{h\in F_{p}}\zeta_{p}^{\frac{f^{*}\left(a\right)+g^{*}\left(b\right)}{z}h^{2}}-1\right)\\ \; & =\left\{\begin{array}{cc} \eta \left(-c\right)\left(p-1\right)\varepsilon_{f}\varepsilon_{g}{\sqrt{p^{*}}}^{2m+s+t+1}, & if\;f^{*}\left(a\right)+g^{*}\left(b\right)=0\\ -\left(\eta \left(f^{*}\left(a\right)+g^{*}\left(b\right)\right)+\eta \left(-c\right)\right)\varepsilon_{f}\varepsilon_{g}{\sqrt{p^{*}}}^{2m+s+t+1}, & if\;f^{*}\left(a\right)+g^{*}\left(b\right)\neq 0 \end{array}\right.\\ \; & =\left\{\begin{array}{cc} \eta \left(-c\right)\left(p-1\right)\varepsilon_{f}\varepsilon_{g}{\sqrt{p^{*}}}^{2m+s+t+1}, & if\;f^{*}\left(a\right)+g^{*}\left(b\right)=0,\\ -2\eta \left(-c\right)\varepsilon_{f}\varepsilon_{g}{\sqrt{p^{*}}}^{2m+s+t+1}, & if\;\eta (f^{*}\left(a\right)+g^{*}\left(b\right))=\eta \left(-c\right),\\ 0, & if\;\eta (f^{*}\left(a\right)+g^{*}\left(b\right))\neq \eta \left(-c\right). \end{array}\right. \end{array}$$

(3) The last case is that $l_{f}=2$ and $l_{g}=p-1$. Then we deduce that
$$\begin{array}{cc} S_{c} & =\varepsilon_{f}\varepsilon_{g}{\sqrt{p^{*}}}^{2m+s+t}\sum_{z\in F_{p}^{*}}\zeta_{p}^{-cz}\eta \left(z\right)\sum_{h\in F_{p}^{*}}\zeta_{p}^{z\left({\left(\frac{h}{z}\right)}^{2}f^{*}\left(a\right)+{\left(\frac{h}{z}\right)}^{p-1}g^{*}\left(b\right)\right)}\\ \; & =\varepsilon_{f}\varepsilon_{g}{\sqrt{p^{*}}}^{2m+s+t}\sum_{z\in F_{p}^{*}}\zeta_{p}^{z(g^{*}\left(b\right)-c)}\eta \left(z\right)\sum_{h\in F_{p}^{*}}\zeta_{p}^{\frac{h^{2}}{z}f^{*}\left(a\right)}\\ \; & =\varepsilon_{f}\varepsilon_{g}{\sqrt{p^{*}}}^{2m+s+t}\sum_{z\in F_{p}^{*}}\zeta_{p}^{z(g^{*}\left(b\right)-c)}\eta \left(z\right)\left(\sum_{h\in F_{p}}\zeta_{p}^{\frac{h^{2}}{z}f^{*}\left(a\right)}-1\right)\\ \; & =\left\{\begin{array}{cc} \left(p-1\right)\varepsilon_{f}\varepsilon_{g}{\sqrt{p^{*}}}^{2m+s+t}\sum_{z\in F_{p}^{*}}\zeta_{p}^{z(g^{*}\left(b\right)-c)}\eta \left(z\right), & if\;f^{*}\left(a\right)=0\\ \varepsilon_{f}\varepsilon_{g}{\sqrt{p^{*}}}^{2m+s+t}\sum_{z\in F_{p}^{*}}\zeta_{p}^{z(g^{*}\left(b\right)-c)}\eta \left(z\right)\left(\eta \left(zf^{*}\left(a\right)\right)\sqrt{p^{*}}-1\right), & if\;f^{*}\left(a\right)\neq 0 \end{array}\right.\\ \; & =\left\{\begin{array}{cc} 0, & if\;f^{*}\left(a\right)=0,g^{*}\left(b\right)=c\\ \left(p-1\right)\eta \left(g^{*}\left(b\right)-c\right)\varepsilon_{f}\varepsilon_{g}{\sqrt{p^{*}}}^{2m+s+t+1}, & if\;f^{*}\left(a\right)=0,g^{*}\left(b\right)\neq c\\ \left(p-1\right)\eta \left(f^{*}\left(a\right)\right)\varepsilon_{f}\varepsilon_{g}{\sqrt{p^{*}}}^{2m+s+t+1}, & if\;f^{*}\left(a\right)\neq 0,g^{*}\left(b\right)=c\\ -\left(\eta \left(f^{*}\left(a\right)\right)+\eta \left(g^{*}\left(b\right)-c\right)\right)\varepsilon_{f}\varepsilon_{g}{\sqrt{p^{*}}}^{2m+s+t+1}, & if\;f^{*}\left(a\right)\neq 0,g^{*}\left(b\right)\neq c \end{array}\right.\\ \; & =\left\{\begin{array}{cc} \left(p-1\right)\varepsilon_{f}\varepsilon_{g}{\sqrt{p^{*}}}^{2m+s+t+1}, & if\;f^{*}\left(a\right)=0,g^{*}\left(b\right)-c\in SQ\\ \; & or\;f^{*}\left(a\right)\in SQ,g^{*}\left(b\right)=c,\\ -\left(p-1\right)\varepsilon_{f}\varepsilon_{g}{\sqrt{p^{*}}}^{2m+s+t+1}, & if\;f^{*}\left(a\right)=0,g^{*}\left(b\right)-c\in NSQ\\ \; & or\;f^{*}\left(a\right)\in NSQ,g^{*}\left(b\right)=c,\\ -2\varepsilon_{f}\varepsilon_{g}{\sqrt{p^{*}}}^{2m+s+t+1}, & if\;f^{*}\left(a\right)\in SQ,g^{*}\left(b\right)-c\in SQ,\\ 2\varepsilon_{f}\varepsilon_{g}{\sqrt{p^{*}}}^{2m+s+t+1}, & if\;f^{*}\left(a\right)\in NSQ,g^{*}\left(b\right)-c\in NSQ,\\ 0, & \mathrm{otherwise}. \end{array}\right. \end{array}$$

So, we obtain the conclusion from (Equation 7), completing the proof. □

### 4.2. Weight Distributions of $C_{D_{f,g}}$

Recall that
(9)$$\begin{array}{c} D_{f,g}=\left\{\left(x,y\right)\in F_{q}^{2}\backslash \left\{\left(0,0\right)\right\}:f\left(x\right)+g\left(y\right)=c\right\}, \end{array}$$
where f,g∈WRPB and $c\in F_{p}^{*}$, and a class of linear codes $C_{D_{f,g}}$ are defined by
(10)$$\begin{array}{c} C_{D_{f,g}}=\left\{c\left(a,b\right)={\left(Tr(ax+by)\right)}_{\left(x,y\right)\in D_{f,g}}:a,b\in F_{q}\right\}. \end{array}$$

The length $n_{c}$ of these linear codes equals the size of $D_{f,g}$. So it is determined by
$$\begin{array}{cc} n_{c} & =\#\left\{\left(x,y\right)\in F_{q}^{2}\backslash \left\{\left(0,0\right)\right\}:f\left(x\right)+g\left(y\right)=c\right\}\\ \; & =\frac{1}{p}\sum_{x,y\in F_{q}}\sum_{z\in F_{p}}{\zeta_{p}}^{z\left(f(x)+g(y)-c\right)}\\ \; & =p^{2m-1}+\frac{1}{p}\sum_{z\in F_{p}^{*}}\zeta_{p}^{-cz}\sum_{x\in F_{q}}{\zeta_{p}}^{zf(x)}\sum_{y\in F_{q}}{\zeta_{p}}^{zg(y)}\\ \; & =p^{2m-1}+\frac{1}{p}\sum_{z\in F_{p}^{*}}\zeta_{p}^{-cz}\sigma_{z}\left({\hat{\chi }}_{f}\left(0\right){\hat{\chi }}_{g}\left(0\right)\right)\\ \; & =p^{2m-1}. \end{array}$$

For the weight distributions of $C_{D_{f,g}}$, where c≠0, we have the following two theorems.

**Theorem 1.** 
*Let s+t be even and $c\in F_{p}^{*}$, the code $C_{D_{f,g}}$ be defined by (Equation 9) and (Equation 10). If $l_{f}=l_{g}=p-1$, then $C_{D_{f,g}}$ is a three-weight $\left[p^{2m-1},2m\right]$ linear code with weight distribution listed in Table 1. If $l_{f}=l_{g}=2$, then $C_{D_{f,g}}$ is a three-weight $\left[p^{2m-1},2m\right]$ linear code with weight distribution listed in Table 2. Otherwise, if $l_{f}=2$ and $l_{g}=p-1$, then $C_{D_{f,g}}$ is a five-weight $\left[p^{2m-1},2m\right]$ linear code with weight distribution listed in Table 3. For abbreviation, we write τ=2m+s+t, γ=2m−s−t, $G_{1}=N_{f}\left(0\right)N_{g}\left(c\right)$ and*

$$\begin{array}{cc} G_{2} & =\frac{p-1}{2}\left(N_{f}\left(i\right)R_{g,SQ}\left(c\right)+N_{f}\left(j\right)R_{g,NSQ}\left(c\right)\right),\\ G_{3} & =\frac{p-1}{2}\left(N_{f}\left(i\right)R_{g,NSQ}\left(c\right)+N_{f}\left(j\right)R_{g,SQ}\left(c\right)\right), \end{array}$$

*where i∈SQ and j∈NSQ.*


**Proof.** For c≠0, the length of $C_{D_{f,g}}$ is $n_{c}=p^{2m-1}$. Let $\left(a,b\right)\in F_{q}^{2}\backslash \left\{\left(0,0\right)\right\}$ and the weight of nonzero codeword c(a,b) be denoted by wt(c(a,b)). Then we obviously have that
$$\mathrm{wt}\left(c\left(a,b\right)\right)=n_{c}-N_{c},$$
where $N_{c}$ is given by Lemma 13. Precisely, when $\left(a,b\right)\notin S_{f}\times S_{g}$, we have
$$\begin{array}{c} \mathrm{wt}\left(c\left(a,b\right)\right)=\left(p-1\right)p^{2m-2}, \end{array}$$
and the number of such codewords is $p^{2m}-p^{2m-s-t}-1$, according to Lemma 2. Furthermore, when $\left(a,b\right)\in S_{f}\times S_{g}$, there are three different cases.The first case is that $l_{f}=l_{g}=p-1$. Then it follows from Lemma 13 that
$$\begin{array}{c} \mathrm{wt}\left(c\left(a,b\right)\right)=\left\{\begin{array}{cc} \left(p-1\right)\left(p^{2m-2}-\left(p-1\right)\varepsilon_{f}\varepsilon_{g}{\sqrt{p^{*}}}^{2(m-2)+s+t}\right), & T(c)\;times,\\ \left(p-1\right)\left(p^{2m-2}+\varepsilon_{f}\varepsilon_{g}{\sqrt{p^{*}}}^{2(m-2)+s+t}\right), & F_{1}\;times, \end{array}\right. \end{array}$$
where $F_{1}=p^{2m-s-t}-T\left(c\right)$, and T(c) is computed in Lemma 10. This gives the weight distribution in Table 1.The second case is that $l_{f}=l_{g}=2$. In this case, it follows from Lemma 13 again that
$$\begin{array}{c} \mathrm{wt}\left(c\left(a,b\right)\right)=\left\{\begin{array}{cc} \left(p-1\right)p^{2m-2}-\left(p+1\right)\varepsilon_{f}\varepsilon_{g}{\sqrt{p^{*}}}^{2(m-2)+s+t}, & T_{SQ}\left(c\right)\;times,\\ \left(p-1\right)\left(p^{2m-2}+\varepsilon_{f}\varepsilon_{g}{\sqrt{p^{*}}}^{2(m-2)+s+t}\right), & F_{2}\;times, \end{array}\right. \end{array}$$
where $F_{2}=p^{2m-s-t}-T_{SQ}\left(c\right)$, and $T_{SQ}\left(c\right)$ is computed in Lemma 11. We thus get the weight distribution in Table 2.Finally, we consider the third case that $l_{f}=2$ and $l_{g}=p-1$. By Lemma 13 again, we have
$$\begin{array}{c} \mathrm{wt}\left(c\left(a,b\right)\right)=\left\{\begin{array}{cc} \left(p-1\right)\left(p^{2m-2}-\left(p-1\right)\varepsilon_{f}\varepsilon_{g}{\sqrt{p^{*}}}^{2(m-2)+s+t}\right), & G_{1}\;times,\\ \left(p-1\right)p^{2m-2}-\left(\eta \left(-1\right)p+1\right)\varepsilon_{f}\varepsilon_{g}{\sqrt{p^{*}}}^{2(m-2)+s+t}, & G_{2}\;times,\\ \left(p-1\right)p^{2m-2}+\left(\eta \left(-1\right)p-1\right)\varepsilon_{f}\varepsilon_{g}{\sqrt{p^{*}}}^{2(m-2)+s+t}, & G_{3}\;times,\\ \left(p-1\right)\left(p^{2m-2}+\varepsilon_{f}\varepsilon_{g}{\sqrt{p^{*}}}^{2(m-2)+s+t}\right), & G_{4}\;times, \end{array}\right. \end{array}$$
where $G_{4}=p^{2m-s-t}-\sum_{i=1}^{3}G_{i}$. The multiplicity of each nonzero weight comes from Lemmas 7 and 9, namely,
$$\begin{array}{cc} G_{1} & =\#\left\{\left(a,b\right)\in S_{f}\times S_{g}:f^{*}\left(a\right)=0,g^{*}\left(b\right)=c\right\}=N_{f}\left(0\right)N_{g}\left(c\right),\\ G_{2} & =\#\{\left(a,b\right)\in S_{f}\times S_{g}:f^{*}\left(a\right)\left(g^{*}\left(b\right)-c\right)\in SQ\}\\ \; & =\frac{p-1}{2}\left(N_{f}\left(i\right)R_{g,SQ}\left(c\right)+N_{f}\left(j\right)R_{g,NSQ}\left(c\right)\right),\\ G_{3} & =\#\{\left(a,b\right)\in S_{f}\times S_{g}:f^{*}\left(a\right)\left(g^{*}\left(b\right)-c\right)\in NSQ\}\\ \; & =\frac{p-1}{2}\left(N_{f}\left(i\right)R_{g,NSQ}\left(c\right)+N_{f}\left(j\right)R_{g,SQ}\left(c\right)\right), \end{array}$$
where i∈SQ and j∈NSQ. The weight distribution is summarized in Table 3. □

**Theorem 2.** 
*Let s+t be odd, $c\in F_{p}^{*}$ and the code $C_{D_{f,g}}$ be defined by (Equation 9) and (Equation 10). If $l_{f}=l_{g}=p-1$, then $C_{D_{f,g}}$ is a three-weight $\left[p^{2m-1},2m\right]$ linear code with weight distribution listed in Table 4. If $l_{f}=l_{g}=2$, then $C_{D_{f,g}}$ is a three-weight $\left[p^{2m-1},2m\right]$ linear code with weight distribution listed in Table 5. If $l_{f}=2$, $l_{g}=p-1$, then $C_{D_{f,g}}$ is a five-weight $\left[p^{2m-1},2m\right]$ linear code with weight distribution listed in Table 6. For briefness, we set τ=2m+s+t, γ=2m−s−t and*

$$\begin{array}{cc} I_{1} & =N_{f}\left(0\right)R_{g,SQ}\left(c\right)+\frac{p-1}{2}N_{f}\left(i\right)N_{g}\left(c\right),\\ I_{2} & =N_{f}\left(0\right)R_{g,NSQ}\left(c\right)+\frac{p-1}{2}N_{f}\left(j\right)N_{g}\left(c\right),\\ I_{3} & =\frac{p-1}{2}N_{f}\left(i\right)R_{g,SQ}\left(c\right),\\ I_{4} & =\frac{p-1}{2}N_{f}\left(j\right)R_{g,NSQ}\left(c\right), \end{array}$$

*where i∈SQ and j∈NSQ.*


**Proof.** Let c≠0 and $\left(a,b\right)\in F_{q}^{2}\backslash \left\{\left(0,0\right)\right\}$. The weight of nonzero codeword c(a,b) is given by
$$\mathrm{wt}\left(c\left(a,b\right)\right)=n_{c}-N_{c},$$
where $n_{c}=p^{2m-1}$ and $N_{c}$ is computed in Lemma 14. According to Lemma 14, when (a,b)≠(0,0), three distinct cases shall be distinguished.For the first case $l_{f}=l_{g}=p-1$, it follows from Lemma 14 that
$$\begin{array}{c} \mathrm{wt}\left(c\left(a,b\right)\right)=\left\{\begin{array}{cc} \left(p-1\right)\left(p^{2m-2}-\varepsilon_{f}\varepsilon_{g}{\sqrt{p^{*}}}^{2m+s+t-3}\right), & V_{SQ}\left(c\right)\;times,\\ \left(p-1\right)\left(p^{2m-2}+\varepsilon_{f}\varepsilon_{g}{\sqrt{p^{*}}}^{2m+s+t-3}\right), & V_{NSQ}\left(c\right)\;times,\\ \left(p-1\right)p^{2m-2}, & F_{3}\;times, \end{array}\right. \end{array}$$
where $F_{3}=p^{2m}-1-V_{SQ}\left(c\right)-V_{NSQ}\left(c\right)$, $V_{SQ}\left(c\right)$ and $V_{NSQ}\left(c\right)$ are computed in Lemma 12. From the above arguments, we obatain the conlusion given in Table 4.For the second case $l_{f}=l_{g}=2$, it follows from Lemma 13 again that
$$\begin{array}{c} \mathrm{wt}\left(c\left(a,b\right)\right)=\left\{\begin{array}{cc} \left(p-1\right)\left(p^{2m-2}-\eta \left(-c\right)\varepsilon_{f}\varepsilon_{g}{\sqrt{p^{*}}}^{2m+s+t-3}\right), & T(0)\;times,\\ \left(p-1\right)p^{2m-2}+2\eta \left(-c\right)\varepsilon_{f}\varepsilon_{g}{\sqrt{p^{*}}}^{2m+s+t-3}, & T_{SQ}\left(-c\right)\;times,\\ \left(p-1\right)p^{2m-2}, & F_{4}\;times, \end{array}\right. \end{array}$$
where $F_{4}=p^{2m}-1-T\left(0\right)-T_{SQ}\left(-c\right)$, and T(0) and $T_{SQ}\left(c\right)$ are computed in Lemmas 10 and 11, respectively. This yields the weight distribution in Table 5.Finally, for the third case $l_{f}=2$ and $l_{g}=p-1$, by Lemma 13 again, we have
$$\begin{array}{c} \mathrm{wt}\left(c\left(a,b\right)\right)=\left\{\begin{array}{cc} \left(p-1\right)\left(p^{2m-2}-\varepsilon_{f}\varepsilon_{g}{\sqrt{p^{*}}}^{2m+s+t-3}\right), & I_{1}\;times,\\ \left(p-1\right)\left(p^{2m-2}+\varepsilon_{f}\varepsilon_{g}{\sqrt{p^{*}}}^{2m+s+t-3}\right), & I_{2}\;times,\\ \left(p-1\right)p^{2m-2}+2\varepsilon_{f}\varepsilon_{g}{\sqrt{p^{*}}}^{2m+s+t-3}, & I_{3}\;times,\\ \left(p-1\right)p^{2m-2}-2\varepsilon_{f}\varepsilon_{g}{\sqrt{p^{*}}}^{2m+s+t-3}, & I_{4}\;times,\\ \left(p-1\right)p^{2m-2}, & I_{5}\;times, \end{array}\right. \end{array}$$
where $I_{5}=p^{2m}-1-\sum_{i=1}^{4}I_{i}$. The multiplicity can be determined from Lemmas 7 and 9, namely,
$$\begin{array}{cc} I_{1} & =\#\{\left(a,b\right)\in S_{f}\times S_{g}:f^{*}\left(a\right)=0,g^{*}\left(b\right)-c\in SQorf^{*}\left(a\right)\in SQ,g^{*}\left(b\right)=c\}\\ & =N_{f}\left(0\right)R_{g,SQ}\left(c\right)+\frac{p-1}{2}N_{f}\left(i\right)N_{g}\left(c\right),\\ I_{2} & =\#\{\left(a,b\right)\in S_{f}\times S_{g}:f^{*}\left(a\right)=0,g^{*}\left(b\right)-c\in NSQorf^{*}\left(a\right)\in NSQ,g^{*}\left(b\right)=c\}\\ & =N_{f}\left(0\right)R_{g,NSQ}\left(c\right)+\frac{p-1}{2}N_{f}\left(j\right)N_{g}\left(c\right),\\ I_{3} & =\#\left\{\left(a,b\right)\in S_{f}\times S_{g}:f^{*}\left(a\right)\in SQ,g^{*}\left(b\right)-c\in SQ\right\}=\frac{p-1}{2}N_{f}\left(i\right)R_{g,SQ}\left(c\right),\\ I_{4} & =\#\{\left(a,b\right)\in S_{f}\times S_{g}:f^{*}\left(a\right)\in NSQ,g^{*}\left(b\right)-c\in NSQ\}\\ \; & =\frac{p-1}{2}N_{f}\left(j\right)R_{g,NSQ}\left(c\right), \end{array}$$
where i∈SQ and j∈NSQ. This gives the weight distribution in Table 6. □

## 5. Minimality of the Codes and Their Applications

In 1979, Shamir [26] and Blakley [27] introduced the notion of secret sharing schemes. Since then, secret sharing schemes have become an important application of linear codes. In recent years, secret sharing schemes have been widely used in cloud environments, banking systems, electronic voting systems and so on.

Any linear code can be employed to construct secret sharing schemes by considering the access structure. However, the access structure based on a linear code is very complicated, and only can be determined in several special cases. One of these cases is that each codeword of the code is minimal.

If a nonzero codeword of a linear code *C* solely covers its scalar multiples, but no other nonzero codewords, then it is called a minimal codeword. The code *C* is said to be minimal if each nonzero codeword of *C* is minimal.

It is naturally difficult to find minimal codes by definition. Fortunately, in 1998, Ashikhmin and Barg [28] provided simple criteria to determine whether a given linear code is minimal.

**Lemma 15** (Ashikhmin-Barg Bound [28]). *Let C be a linear code over $F_{p}$. Then all nonzero codewords of C are minimal, provided that*
$$\frac{w_{min}}{w_{max}}>\frac{p-1}{p},$$
*where $w_{min}$ and $w_{max}$ stand for the minimum and maximum nonzero weights in C, respectively.*

Now we will show under what circumstances the linear codes constructed in this paper are minimal. The following theorem is verified directly according to Lemma 15.

**Theorem 3.** 
*When 2m−(s+t)⩾4 with any odd prime p, the linear codes described in Table 1, Table 2, Table 3, Table 4, Table 5 and Table 6 are minimal.*


Under the framework [29], the minimal codes in Theorem 3 can be applied to construct secret sharing schemes with good access structures. An example is showed in detail in the following.

**Theorem 4** (Proposition 2, [29]). *Let C be an [n,k] code over $F_{q}$, and let $G=[g_{0},g_{1},\cdots ,g_{n-1}]$ be its generator matrix. If C is minimal, then in the secret sharing schemes based on the dual code $C^{⊥}$, there are altogether $q^{k-1}$ minimal access sets. In addition, we have the following assertions.*

(1)
*If $g_{i}$ is a multiple of $g_{0}$, 1≤i≤n−1, then participant $P_{i}$ must be in every minimal access set. Such a participant is called a dictatorial participant.*
(2)
*If $g_{i}$ is not a multiple of $g_{0}$, 1≤i≤n−1, then participant $P_{i}$ must be in $\left(q-1\right)q^{k-2}$ out of $q^{k-1}$ minimal access sets.*


Now, we take the code $C_{D_{f,g}}$ described in Table 1 as an example. If we take p=5, m=4, s+t=4 and $\epsilon_{f}\epsilon_{g}=1$, then $C_{D_{f,g}}$ has length n=78125 and dimension k=8. From Table 1, the weight enumerator of $C_{D_{f,g}}$ is $1+120z^{52500}+389999z^{62500}+505z^{65000}$. The code $C_{D_{f,g}}$ is minimal due to Theorem 3. Note that the minimum distance of its dual code is $d^{⊥}=2$. According to Theorem 4, we get the following theorem.

**Theorem 5.** 
*Let p=5, m=4, s+t=4 and $\epsilon_{f}\epsilon_{g}=1$ and $G=[g_{0},g_{1},\cdots ,g_{78124}]$ be the generator matrix of the code $C_{D_{f,g}}$ described in Table 1. Then in the secret sharing scheme based on the dual code $C_{D_{f,g}}^{⊥}$, there are altogether $5^{7}$ minimal access sets. In addition, we have the following assertions.*


(1)
*If $g_{i}$ is a multiple of $g_{0}$, 1≤i≤78124, then participant $P_{i}$ must be in every minimal access set and $P_{i}$ is a dictatorial participant.*
(2)
*If $g_{i}$ is not a multiple of $g_{0}$, 1≤i≤78124, then participant $P_{i}$ must be in $4\times 5^{6}$ out of $5^{7}$ minimal access sets.*


## 6. Conclusions

The paper studied the construction of linear codes from two weakly regular *s*-plateaued and *t*-plateaued balanced functions. Hence, this was an extension of the results in [2] and [10]. Additionally, because of the minimality, the codes we constructed are suitable for secret sharing schemes. However, no one finds an example of weakly regular plateaued balanced functions in the set WRPB. It would be desirable to find such a function, but we have not been able to do this.

## Figures and Tables

**Table 1 entropy-25-00369-t001:** The weight distribution of $C_{D_{f,g}}$ when $l_{f}=l_{g}=p-1$ and 2|s+t.

Weight	Multiplicity
0	1
$\left(p-1\right)p^{2m-2}$	$p^{2m}-p^{\gamma }-1$
$\left(p-1\right)(p^{2m-2}-\left(p-1\right)\varepsilon_{f}\varepsilon_{g}{\sqrt{p^{*}}}^{\tau -4})$	$p^{\gamma -1}-p^{-1}\varepsilon_{f}\varepsilon_{g}{\sqrt{p^{*}}}^{\gamma }$
$\left(p-1\right)(p^{2m-2}+\varepsilon_{f}\varepsilon_{g}{\sqrt{p^{*}}}^{\tau -4})$	$\left(p-1\right)p^{\gamma -1}+p^{-1}\varepsilon_{f}\varepsilon_{g}{\sqrt{p^{*}}}^{\gamma }$

**Table 2 entropy-25-00369-t002:** The weight distribution of $C_{D_{f,g}}$ when $l_{f}=l_{g}=2$ and 2|s+t.

Weight	Multiplicity
0	1
$\left(p-1\right)p^{2m-2}$	$p^{2m}-p^{\gamma }-1\;$
$\left(p-1\right)p^{2m-2}-\left(p+1\right)\varepsilon_{f}\varepsilon_{g}{\sqrt{p^{*}}}^{\tau -4}$	$\frac{p-1}{2}(p^{\gamma -1}-p^{-1}\varepsilon_{f}\varepsilon_{g}{\sqrt{p^{*}}}^{\gamma })$
$\left(p-1\right)\left(p^{2m-2}+\varepsilon_{f}\varepsilon_{g}{\sqrt{p^{*}}}^{\tau -4}\right)$	$\frac{p+1}{2}p^{\gamma -1}+\frac{p-1}{2p}\varepsilon_{f}\varepsilon_{g}{\sqrt{p^{*}}}^{\gamma }$

**Table 3 entropy-25-00369-t003:** The weight distribution of $C_{D_{f,g}}$ when $l_{f}=2$, $l_{g}=p-1$ and 2|s+t.

Weight	Multiplicity
0	1
$\left(p-1\right)p^{2m-2}$	$p^{2m}-p^{\gamma }-1\;$
$\left(p-1\right)(p^{2m-2}-\left(p-1\right)\varepsilon_{f}\varepsilon_{g}{\sqrt{p^{*}}}^{\tau -4})$	$G_{1}$
$\left(p-1\right)p^{2m-2}-\left(\eta \left(-1\right)p+1\right)\varepsilon_{f}\varepsilon_{g}{\sqrt{p^{*}}}^{\tau -4}$	$G_{2}$
$\left(p-1\right)p^{2m-2}+\left(\eta \left(-1\right)p-1\right)\varepsilon_{f}\varepsilon_{g}{\sqrt{p^{*}}}^{\tau -4}$	$G_{3}$
$\left(p-1\right)(p^{2m-2}+\varepsilon_{f}\varepsilon_{g}{\sqrt{p^{*}}}^{\tau -4})$	$p^{\gamma }-G_{1}-G_{2}-G_{3}$

**Table 4 entropy-25-00369-t004:** The weight distribution of $C_{D_{f,g}}$ when $l_{f}=l_{g}=p-1$ and 2∤s+t.

Weight	Multiplicity
0	1
$\left(p-1\right)(p^{2m-2}-\varepsilon_{f}\varepsilon_{g}{\sqrt{p^{*}}}^{\tau -3})$	$\frac{p-1}{2}p^{\gamma -1}-\frac{1+\eta (c)}{2}\varepsilon_{f}\varepsilon_{g}{\sqrt{p^{*}}}^{\gamma -1}$
$\left(p-1\right)(p^{2m-2}+\varepsilon_{f}\varepsilon_{g}{\sqrt{p^{*}}}^{\tau -3})$	$\frac{p-1}{2}p^{\gamma -1}+\frac{1-\eta (c)}{2}\varepsilon_{f}\varepsilon_{g}{\sqrt{p^{*}}}^{\gamma -1}$
$\left(p-1\right)p^{2m-2}$	$p^{2m}-1-\left(p-1\right)p^{\gamma -1}+\eta \left(c\right)\varepsilon_{f}\varepsilon_{g}{\sqrt{p^{*}}}^{\gamma -1}$

**Table 5 entropy-25-00369-t005:** The weight distribution of $C_{D_{f,g}}$ when $l_{f}=l_{g}=2$ and 2∤s+t.

Weight	Multiplicity
0	1
$\left(p-1\right)(p^{2m-2}-\eta \left(-c\right)\varepsilon_{f}\varepsilon_{g}{\sqrt{p^{*}}}^{\tau -3})$	$p^{\gamma -1}$
$\left(p-1\right)p^{2m-2}+2\eta \left(-c\right)\varepsilon_{f}\varepsilon_{g}{\sqrt{p^{*}}}^{\tau -3}$	$\frac{p-1}{2}(p^{\gamma -1}+\varepsilon_{f}\varepsilon_{g}\eta \left(-c\right){\sqrt{p^{*}}}^{\gamma -1})$
$\left(p-1\right)p^{2m-2}$	$p^{2m}-1-\frac{p+1}{2}p^{\gamma -1}-\frac{p-1}{2}\varepsilon_{f}\varepsilon_{g}\eta \left(-c\right){\sqrt{p^{*}}}^{\gamma -1}$

**Table 6 entropy-25-00369-t006:** The weight distribution of $C_{D_{f,g}}$ when $l_{f}=2$, $l_{g}=p-1$ and 2∤s+t.

Weight	Multiplicity
0	1
$\left(p-1\right)(p^{2m-2}-\varepsilon_{f}\varepsilon_{g}{\sqrt{p^{*}}}^{\tau -3})$	$I_{1}$
$\left(p-1\right)(p^{2m-2}+\varepsilon_{f}\varepsilon_{g}{\sqrt{p^{*}}}^{\tau -3})$	$I_{2}$
$\left(p-1\right)p^{2m-2}+2\varepsilon_{f}\varepsilon_{g}{\sqrt{p^{*}}}^{\tau -3}$	$I_{3}$
$\left(p-1\right)p^{2m-2}-2\varepsilon_{f}\varepsilon_{g}{\sqrt{p^{*}}}^{\tau -3}$	$I_{4}$
$\left(p-1\right)p^{2m-2}$	$p^{2m}-1-I_{1}-I_{2}-I_{3}-I_{4}$

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
