# Peer review of "Linear Codes from Two Weakly Regular Plateaued Balanced Functions"

_entropy, 2023, doi:10.3390/e25020369_

Round 1

Reviewer 1 Report

I believe this paper would be much better suited to be considered by a mathematical journal, such as for references [17] and [18], which are the ones most related to this work. The authors do not propose new codes. Rather, they study the weight distributions of (linear) codes that would be obtained using weakly regular plateaued balance functions as a starting point. However, no such functions are identified, and as the authors mention in the conclusion, they were not able to identify any such function. Thus, the paper is purely theoretical, completely based on mathematical derivations and more appropriate for a mathematical journal. I believe most readers of Entropy, including most of the readers working in the area of coding, will not be able to follow the paper, which is presented assuming that the reader is very familiar with the necessary background (something that regular readers will not have). 

Author Response

This comment is very important. Entropy is an international and interdisciplinary journal of entropy and information studies with strong inclusiveness, which can provide readers with more and broader reading resources. That's why we chose entropy in the first place. Our research work is based on the information theory of coding, and it is practical to use some mathematical derivation methods to realize the proof of coding theory. This is just to give the reader a better understanding of how information theory is being realized step by step. Here, mathematics is not our research purpose. In this work, our goal is how to construct a linear code family with up to five non-zero weights (and check their minimality), so that the constructed code is more useful in secret sharing schemes. 

Reviewer 2 Report

The paper proposed a new construction of linear codes with few weights from defining sets. Such codes utilized two weakly regular s-plateaued and t-plateaued balanced functions and can be applied for secret sharing schemes. The only concern is that there are no examples of weakly regular plateaued balanced functions. The authors state it as an open question. Before manuscript acceptance, I would ask the authors to incorporate the following changes:

1. You fixed p to be an odd prime, but what about the binary case? Note that from the practical point of view, the binary case is more interesting. 

2. Please add recent papers on constructions of codes by defining sets, for example, "Application of optimal p-ary linear codes to alphabet-optimal locally repairable codes" 

3. Please include necessary definitions, for example 

        Trace function

        Extended field

       s-plateaued balanced function

4. Please also explain the abbreviation at the first time of appearance, for example

        SQ

       NSQ

      WRPB

      WRP

5. Please add a reference to Walsh transform

6. Last lines of Lemma 11 and Lemma 12 need to be explained

7. The first sentence after lemma 2 appears two consecutive times

8. What are the parameters of secret sharing schemes that can be obtained by your codes?

Author Response

Thank you for your comments. Please see the attached file for our response.

Reviewer 3 Report

see the attached file.

Author Response

Thank you so much. Please see the PDF file for our response.

Round 2

Reviewer 2 Report

The authors incorporated all my comments, and I recommend the acceptance of the paper in its current form. 

Author Response

Thanks a lot for your useful comments.

Reviewer 3 Report

The reviewer would like to thank the authors for the revised paper and the response letter. My comments about them are listed below. 

- The balanced quadratic functions are in the set of WRPB. So, the authors can easily find concrete examples of quadratic balanced functions for Theorems 1 and 2.

- The authors can give in the paper Theorems 1 and 2 that were given in the response letter of Report 2.

The topic of this paper is covered by my research field.  The proposed results are valuable in the literature, and they are correct as far as checked.  However,  the proposed results (two linear codes obtained from balanced functions) are not sufficient for this high-level journal. If more linear codes are constructed from unbalanced functions,  then this paper will have a high scientific level, and it can be published in this journal. Otherwise, I think that the scientific level of the current results is weak for this journal. Finally, I leave the final decision to the editor about this issue.  
